**Data Availability Statement:** Data cannot be shared publicly due to potentially identifying and sensitive patient information. Data are only available with approval of the Institutional Ethics

# The clinicopathological significance of SWI/SNF alterations in gastric cancer is associated with the molecular subtypes

Shih-Chiang Huang[1,2], Kwai-Fong Ng[1], Ian Yi-Feng Chang[3], Chee-Jen Chang[2,4,5,6,7], Yi-Chun Chao[1], Shu-Chen Chang[4], Min-Chi Chen[8,9], Ta-Sen Yeh[10], Tse-Ching Chen[1]*

**1** Department of Anatomic Pathology, Linkou Chang Gung Memorial Hospital, Chang Gung University, College of Medicine, Taoyuan, Taiwan, **2** Graduate Institute of Clinical Medical Sciences, College of Medicine, Chang Gung University, Taoyuan, Taiwan, **3** Molecular Medicine Research Center, Chang Gung University, Taoyuan, Taiwan, **4** Research Services Center for Health Information, Chang Gung University, Taoyuan, Taiwan, **5** Clinical Informatics and Medical Statistics Research Center, Chang Gung University, Taoyuan, Taiwan, **6** Department of Biomedical Sciences, Chang Gung University, Taoyuan, Taiwan, **7** Department of Cardiology, Linkou Chang Gung Memorial Hospital, Chang Gung University, College of Medicine, Taoyuan, Taiwan, **8** Department of Public Health, Biostatistics Consulting Center, College of Medicine, Chang Gung University, Taoyuan, Taiwan, **9** Department of Hematology and Oncology, Chiayi Chang Gung Memorial Hospital, Chang Gung University, College of Medicine, Chiayi, Taiwan, **10** Department of Surgery, Linkou Chang Gung Memorial Hospital, Chang Gung University, College of Medicine, Taoyuan, Taiwan

* ctc323@cgmh.org.tw

## Abstract

The clinicopathological significance of altered SWI/SNF complex has not been well evaluated in gastric cancer (GC). We examined SMARCA2, SMARCA4, SMARCB1 and ARID1A expression by immunohistochemistry in 1224 surgically resected GCs with subtyping into Epstein-Barr virus (EBV), microsatellite instability (MSI) and non-EBV/MSI Lauren histotypes. SWI/SNF mutations were investigated using the GC dataset of the TCGA Pan-Cancer Atlas. Clinicopathological association was assessed by statistical analysis. There were 427 cases (35%) of SWI/SNF-attenuated GC, including 344 SMARCA2 (28%), 28 SMARCA4 (2%), 11 SMARCB1 (1%) and 197 ARID1A (16%) cases. Simultaneous alterations of multiple subunits were observed. Compared to SWI/SNF-retained cases, SWI/SNF-attenuated GC exhibited a significant predilection to older ages, EBV and MSI genotypes, higher lymphatic invasion and less hematogenous recurrence (*P* < 0.05). SWI/SNF attenuation was an independent risk factor for short overall survival (*P* = 0.001, hazard ratio 1.360, 95% confidence interval 1.138–1.625). The survival impact stemmed from SMARCA2-attenuated GCs in stage III and non-EBV/MSI diffuse/mixed subtypes (*P* = 0.019 and < 0.001, respectively). ARID1A-lost/heterogeneous GCs were more aggressive in the EBV genotype (*P* = 0.016). SMARCB1 or SMARCA4 loss was not restricted to rhabdoid/undifferentiated carcinoma. In the TCGA dataset, 223 of 434 GCs (52%) harbored deleterious SWI/SNF mutations, including *ARID1A* (27%), *SMARCA2* (9%), *ARID2* (9%), *ARID1B* (8%), *PBRM1* (7%), and *SMARCA4* (7%). SWI/SNF-mutated GCs displayed a favorable outcome owing to the high percentage with the MSI genotype. In conclusion, SWI/

Committee of the Chang Gung Memorial Hospital (irb1@cgmh.org.tw) or via the corresponding author (ctc323@cgmh.org.tw) for researchers who meet the criteria for access to confidential data.

**Funding:** This work was supported by grants from the Ministry of Science and Technology (108-2320-B-182A-018, 106-2320-B-182A-011-MY3 and 105-2320-B-182A-014) and the Chang Gung Memorial Hospital (CMRPG3F2073, CMRPG5J0091, CIRPG3D0153, CMRP3C1323 and CMRPG3G0553). The funders had no role in study design, data collection and analysis, decision to publish, or preparation of the manuscript. All the funding or sources of support were received during this study and there was no additional external funding received for this study.

**Competing interests:** The authors declare that they have no competing interests.

**Abbreviations:** GC, gastric cancer; TCGA, The Cancer Genome Atlas; EBV, Epstein-Barr virus; MSI, microsatellite instability; GS, genomically stable; CIN, chromosomal instability; EBER-ISH, EBV-encoded small RNA in situ hybridization; MMR-IHC, immunohistochemistry of DNA mismatch repair proteins; SWI/SNF, Switch/sucrose non-fermentable complex; CNA, copy number alteration; STAD, stomach adenocarcinoma; PBAF, Polybromo-associated BRG1/BRM-associated factor; ncBAF, non-canonical BAF.

SNF-altered GCs are common and the clinicopathological significance is related to the genotype.

## Introduction

Gastric cancer (GC) continues to be ranked third in cancer-related mortality worldwide [1]. Recently, molecular knowledge regarding gastric carcinogenesis progresses dramatically. The Cancer Genome Atlas (TCGA) network used whole genome approaches to divide GC into Epstein-Barr virus (EBV)-positive, microsatellite instability (MSI)-high, genomically stable (GS) and chromosomal instability (CIN) subtypes [2]. In our previous work, we integrated EBV-encoded small RNA in situ hybridization (EBER-ISH), immunohistochemistry of DNA mismatch repair proteins (MMR-IHC) and Lauren histotyping to design a practical GC sub-typing algorithm, parallel to the TCGA classification [3]. In brief, the Lauren intestinal and dif-fuse/mixed division was done after EBV and MSI-associated GCs were subtracted. The non-EBV/MSI intestinal and diffuse/mixed subtypes had clinical and molecular similarity to the TCGA CIN and GS variants, respectively [3].

The other next-generation sequencing studies have further unveiled new and prevailing genetic mutations. Of note, *ARID1A* (AT-rich interactive domain 1A) mutations have emerged in approximately 10% of GCs and were enriched in EBV or MSI-associated subtypes [4]. ARID1A is a member of the SWI/SNF (SWItch/Sucrose Non-Fermentable) complex that regulates chromatin remodeling, thereby controlling genomic transcription. The SWI/SNF complex is a multiprotein assembly, consisting of ATPase (SMARCA2, SMARCA4), core sub-units (SMARCB1, SMARCC1, SMARCC2) and variant subunits (ARID1A/B, ARID2, PBRM1, etc.). Around 20% of all human cancers harbor mutations affecting the SWI/SNF complex, implicating the pivotal role of chromatin remodelers in tumorigenesis [5]. In spite of several studies investigating ARID1A alterations in GC [6], data regarding other SWI/SNF subunits are relatively sparse. Our previous study identified altered SMARCA4 expression in 2% of GCs, and SMARCA4-altered GC exhibited intratumoral heterogeneity, histomorpholo-gical diversity and prognostic significance in EBV-associated and non-EBV/MSI intestinal subtypes [7]. Although decreased SMARCA2 expression has been described in GC, the associ-ation of GC molecular subtypes is unknown [8].

As SWI/SNF-targeted agents are emerging [9, 10], we plan to explore the SWI/SNF alterations on the current viewpoint of GC molecular heterogeneity by using a cohort of 1224 patients, which have been subtyped into EBV, MSI, and non-EBV/MSI Lauren histotypes in our previous study [3]. The reasons for selecting these 4 SWI/SNF subunits (SMARCA2, SMARCA4, SMARCB1, ARID1A) are that SMARCA2/4 are the most critical ATPase subunits, SMARCB1 is the core subunit linked to undifferentiated/rhabdoid tumors, and ARID1A is the most mutated variant subunit. Since the members of SWI/SNF subunits are increasingly recognized [11], we also investigated SWI/SNF mutations and copy number alterations (CNAs) using the stomach adeno-carcinoma (STAD) dataset of the TCGA Pan-Cancer Atlas [12]. Through combinatorial immu-nohistochemical and genomic analysis, we anticipate providing the clinicopathologic significance of SWI/SNF-altered GCs and the association with molecular subtypes.

## Materials and methods

### Case collection

We enrolled 1,224 patients who received gastrectomy for GC between January 1999 and December 2007 from the archive of the Department of Anatomic Pathology at Linkou Chang

Gung Memorial Hospital in Taiwan. Patient demographics, tumor characteristics and clinical outcomes were collected from the medical records and the Taiwan Cancer Registry database. Patient survival was traced through July 31, 2018. All data were anonymized by symbols when we accessed them. This study was approved by the institutional review board at our hospital.

### Tissue microarrays, EBER-ISH, IHC and HER2 testing

Data regarding EBV, MSI, HER2 and SMARCA4 have been reported in our previous publications [3, 7, 13, 14]. Briefly, we constructed tissue microarrays using an automated tissue arrayer (BEECHER ATA-27, Beecher Instruments, Sun Prairie, WI, USA). Tissue sections were subjected to EBER-ISH and MMR-IHC(MLH1, MSH2, MSH6, PMS2), HER2, SMARCA2 (HPA029981, 1:50, Sigma-Aldrich, St. Louis, MO), SMARCA4 (EPNCIR111A, 1:50, Abcam), SMARCB1 (25/BAF47, 1:50, BD Biosciences, San Jose, CA) and ARID1A (EPR13501, 1:50, Abcam). The procedures were conducted in an automated immunostaining machine (BOND-MAX, Leica Microsystems) with optimal negative and positive controls, according to the manufacturer's protocols. In this study, GCs were first divided into EBV and MSI, and the remaining negative cases were classified into Lauren intestinal and diffuse/mixed subtypes [3]. According to our previous study, the non-EBV/MSI intestinal and diffuse/mixed subtypes approximately represented TCGA's CIN and GS molecular categories, respectively. HER2 status was determined according to Hofmann's scoring system [15]. Validation of whole tissue sections was performed for cases with attenuated expression of SWI/SNF subunits.

IHC patterns of the SWI/SNF subunits were evaluated according to previous studies [7, 16, 17]. Compared to positive controls with normal epithelial, inflammatory, and fibroblastic cells with uniform and strong expression of the SWI/SNF subunit proteins in their nuclei, cases were categorized as "retained" if the staining intensity was similar to that in normal cells, "reduced" if the staining intensity was substantially weaker or faint but was recognizable, and "lost" if the nuclear staining was completely absent (Fig 1A). Samples with lost or reduced expression in only part of the tumor were designated as "heterogeneous". In this study, we designated all cases with abnormal SWI/SNF expression as SWI/SNF-attenuated GC.

### Statistical analysis

Statistical analysis was conducted using the SPSS software platform (version 20; IBM, New York, NY) and described in our previous study [3]. For variables with $P < 0.2$ by univariate analysis, the multivariate logistic regression model was adopted to clarify the independent factor for attenuated SWI/SNF status. The Cox proportional hazard regression model using backward elimination was performed to identify independent prognostic factors. For the clinicopathologic factors appearing significant in multivariate regression analysis, we progressed to perform subgroup analysis to determine which subgroup was more susceptible for the alterations of SWI/SNI component. Among the most significant independent factors for survival ($P \leq 0.001$; gastrectomy and lymphadenectomy type, combined classification, AJCC stages, chemotherapy treatment), combined classification and AJCC stages likely represented tumor biologic properties and other factors indicated clinical interventions. The regression proportional hazards analysis for interaction $P$ value was done only for AJCC stage since the incidence of SWI/SNF alterations was significantly related to the combined classification ($P < 0.001$), leading to the multicollinear problem. The interaction test for AJCC stage showed significant ($P = 0.04$), and the subgroup analysis was done thereafter.

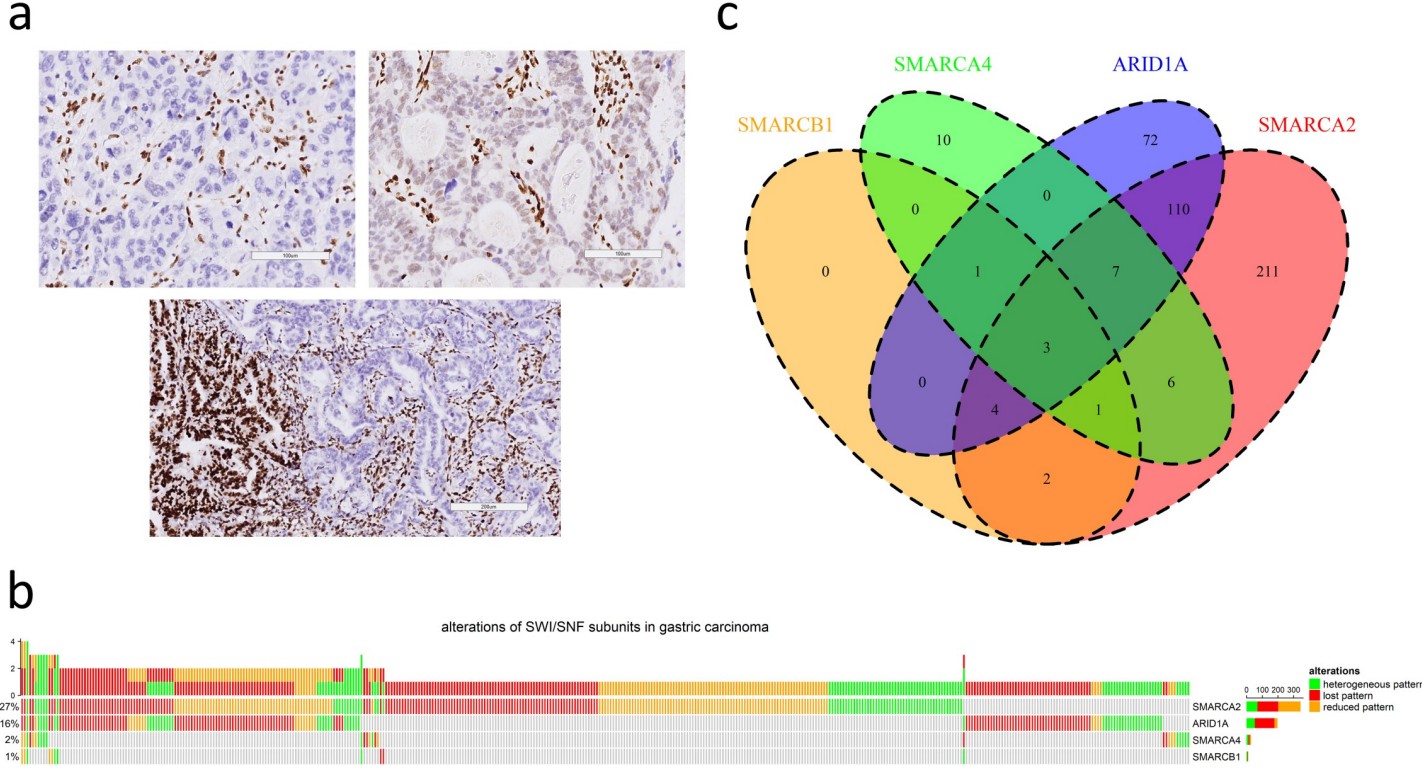

**Fig 1. a.** Three immunohistochemical patterns of attenuated SWI/SNF subunits in gastric cancers (upper left: lost pattern, upper right: reduced pattern, lower: heterogeneous pattern; scale bar in upper left and upper right: 100 μm, scale bar in lower: 200 μm). **b.** The oncoprint plot summarizes the distribution of attenuated SWI/ SNF subunits, including SMARCA2, ARID1A, SMARCA4 and SMARCB1. **c.** The Venn diagram demonstrates synchronous alterations in multiple SWI/SNF subunits.

## TCGA dataset retrieval

We downloaded and analyzed clinical information, somatic variants, and CNAs for 434 STAD patients from the TCGA Pan-Cancer Atlas dataset via cBioPortal (http://download.cbioportal. org/stad_tcga_pan_can_atlas_2018.tar.gz) on June 1, 2020 [18]. The molecular data of all 30 genes encoding SWI/SNF subunits, were explored [11].

## Results

### SWI/SNF-attenuated GC

Among 1224 cases, attenuated SMARCA2, SMARCA4, SMARCB1 and ARID1A expression was observed in 344 (28%), 28 (2%), 11 (1%) and 197 (16%) cases, respectively (Fig 1B). The proportions of lost, reduced and heterogeneous patterns varied among SMARCA2 (131, 39%; 143, 43%; 70, 21%), SMARCA4 (7, 25%; 9, 32%; 12, 43%), SMARCB1 (2, 18%; 4, 36%; 5, 45%) and ARID1A (125, 63%; 20, 10%; 52, 26%). In summary, there were 427 cases (35%) of SWI/ SNF-attenuated GC, and synchronous alterations of multiple SWI/SNF subunits existed in 134 cases (31%) (Fig 1C). The simultaneous attenuation of SMARCA2 and ARID1A expression was most frequent (n = 124).

Compared to the SWI/SNF-retained group, SWI/SNF-attenuated GCs showed a significant predisposition to older patients (Age > 65 years, 56% versus 49%, $P = 0.021$), EBV and MSI genotypes (10% and 15% versus 3% and 7%, $P < 0.001$), patients with lymphatic invasion (63% versus 54%, $P = 0.003$) and patients without hematogenous recurrence (recurrence in the form of visceral metastasis, 29% versus 37%, $P = 0.049$) (Table 1). Remarkably, the SWI/SNF-

**Table 1. Clinicopathological differences between SWI/SNF-retained and SWI/SNF-attenuated gastric cancers.**

| Parameters | Total (n = 1224) | SWI/SNF-retained (n = 797) | SWI/SNF-attenuated (n = 427) | P value |
|---|---|---|---|---|
| Age (median ± SD, yrs) | 66.00 ± 13.65 | 65.00 ± 13.75 | 67.00 ± 13.36 | 0.011 |
| ≤ 65 | 594 (48.5) | 406 (50.9) | 188 (44.0) | 0.021 |
| > 65 | 630 (51.5) | 391 (49.1) | 239 (56.0) | |
| Sex | | | | 0.481 |
| Male | 773 (63.2) | 509 (63.9) | 264 (61.8) | |
| Female | 451 (36.8) | 288 (36.1) | 163 (38.2) | |
| Gastrectomy | | | | 0.016 |
| Proximal/Subtotal | 847 (69.2) | 570 (71.5) | 277 (64.9) | |
| Total | 377 (30.8) | 227 (28.5) | 150 (35.1) | |
| Lymphadenectomy | | | | 0.780 |
| D1/D1+ | 301 (24.6) | 198 (24.8) | 103 (24.1) | |
| D2 | 923 (75.4) | 599 (75.2) | 324 (75.9) | |
| Stump cancer | | | | 0.129 |
| Yes | 59 (4.8) | 33 (4.1) | 26 (6.1) | |
| No | 1165 (95.2) | 764 (95.9) | 401 (93.9) | |
| Localization | | | | 0.209 |
| Upper | 212 (17.3) | 127 (15.9) | 85 (19.9) | |
| Middle | 223 (18.2) | 140 (17.6) | 83 (19.4) | |
| Lower | 743 (60.7) | 499 (62.6) | 244 (57.1) | |
| Diffuse | 46 (3.8) | 31 (3.9) | 15 (1.2) | |
| Size (median ± SD, cm) | 4.0 ± 3.60 | 4.0 ± 3.73 | 4.5 ± 3.36 | 0.042 |
| ≤ 5 | 774 (63.2) | 515 (66.5) | 259 (60.7) | 0.171 |
| > 5 | 450 (36.8) | 282 (35.4) | 168 (39.3) | |
| Differentiation | | | | 0.102 |
| WD/MD | 465 (38.0) | 316 (39.6) | 149 (34.9) | |
| PD | 759 (62.0) | 481 (60.4) | 278 (65.1) | |
| Lauren's classification | | | | 0.950 |
| Intestinal | 592 (48.4) | 386 (48.4) | 206 (48.2) | |
| Diffuse/Mixed | 632 (51.6) | 411 (51.6) | 221 (51.8) | |
| Genotypes[a] | | | | <0.001 |
| EBV | 65 (5.5) | 23 (3.0) | 42 (10.3) | |
| MSI | 114 (9.7) | 54 (7.0) | 60 (14.7) | |
| Intestinal | 467 (39.7) | 327 (42.6) | 140 (34.2) | |
| Diffuse/Mixed | 530 (45.1) | 363 (47.3) | 167 (40.8) | |
| Depth of invasion | | | | 0.060 |
| T1 | 202 (16.5) | 148 (18.6) | 54 (12.6) | |
| T2 | 161 (13.2) | 99 (12.4) | 62 (14.5) | |
| T3 | 280 (22.9) | 179 (22.5) | 101 (23.7) | |
| T4 | 581 (47.5) | 371 (46.5) | 210 (49.2) | |
| Nodal status | | | | 0.146 |
| N0 | 414 (33.8) | 283 (35.5) | 131 (31.6) | |
| N1 | 158 (12.9) | 100 (12.5) | 58 (13.6) | |
| N2 | 208 (17.0) | 141 (17.7) | 67 (15.7) | |
| N3 | 444 (36.3) | 273 (34.3) | 171 (38.5) | |
| LN ratio, median ± SD | 0.14 ± 0.30 | 0.13 ± 0.29 | 0.15 ± 0.31 | 0.081 |
| Distant metastasis | | | | 0.522 |
| M0 | 1109 (90.6) | 719 (90.2) | 390 (91.3) | |

*(Continued)*

**Table 1.** (Continued)

| Parameters | Total (n = 1224) | SWI/SNF-retained (n = 797) | SWI/SNF-attenuated (n = 427) | *P* value |
|---|---|---|---|---|
| M1 | 115 (9.4) | 78 (9.8) | 37 (8.7) | |
| Stage | | | | 0.083 |
| I | 275 (22.5) | 195 (24.5) | 80 (18.7) | |
| II | 246 (20.1) | 151 (18.9) | 95 (22.2) | |
| III | 588 (48.0) | 373 (46.8) | 215 (50.4) | |
| IV | 115 (9.4) | 78 (9.8) | 37 (8.7) | |
| Resection margins | | | | 0.230 |
| Negative | 1090 (89.1) | 716 (89.8) | 374 (87.6) | |
| Positive | 134 (10.9) | 81 (10.2) | 53 (12.4) | |
| Lymphatic invasion[a] | | | | 0.003 |
| No | 513 (42.6) | 358 (45.8) | 155 (36.8) | |
| Yes | 690 (57.4) | 424 (54.2) | 266 (63.2) | |
| Vascular invasion[a] | | | | 0.969 |
| No | 1008 (84.3) | 658 (84.3) | 350 (84.3) | |
| Yes | 188 (15.7) | 123 (15.7) | 65 (15.7) | |
| Perineural invasion[a] | | | | 0.910 |
| No | 563 (47.0) | 365 (46.9) | 198 (47.3) | |
| Yes | 634 (53.0) | 413 (53.1) | 221 (52.7) | |
| HER2 status[a] | | | | 0.079 |
| Negative | 853 (93.5) | 555 (92.5) | 298 (95.5) | |
| Positive | 59 (6.5) | 45 (7.5) | 14 (4.5) | |
| Locoregional recurrence[b] | | | | 0.645 |
| Negative | 354 (67.9) | 228 (67.3) | 126 (69.2) | |
| Positive | 167 (32.1) | 111 (32.7) | 56 (30.8) | |
| Peritoneal recurrence[b] | | | | 0.157 |
| Negative | 322 (61.8) | 217 (64.0) | 105 (57.7) | |
| Positive | 199 (38.2) | 122 (36.0) | 77 (42.3) | |
| Hematogenous recurrence[b] | | | | 0.049 |
| Negative | 343 (65.8) | 213 (62.8) | 130 (71.4) | |
| Positive | 178 (34.2) | 126 (37.2) | 52 (28.6) | |
| Lymph node recurrence[b] | | | | 0.201 |
| Negative | 414 (79.5) | 275 (81.1) | 139 (76.4) | |
| Positive | 107 (20.5) | 64 (18.9) | 43 (23.6) | |
| Chemotherapy[c] | | | | 0.032 |
| Negative | 243 (25.7) | 140 (23.4) | 103 (29.7) | |
| Positive | 703 (74.3) | 459 (76.6) | 244 (70.3) | |

Figures are numbers with percentages in parentheses.

EBV, Epstein-Barr virus; MSI, microsatellite instability; SD, standard deviation; WD/MD, well differentiated/moderately differentiated; PD, poorly differentiated; LN ratio, ratio of metastatic to retrieved lymph nodes.

[a] Not all data were available.

[b] Stage I-III cases with available data regarding recurrence site.

[c] Stage II-IV cases with available data of chemotherapy.

attenuated group received more often received a total gastrectomy (35% versus 29%, *P* = 0.016) but less chemotherapy (70% versus 76%, *P* = 0.032), indicating the survival of patients with SWI/SNF-attenuated GC might have substantial bias (see below). Trends for

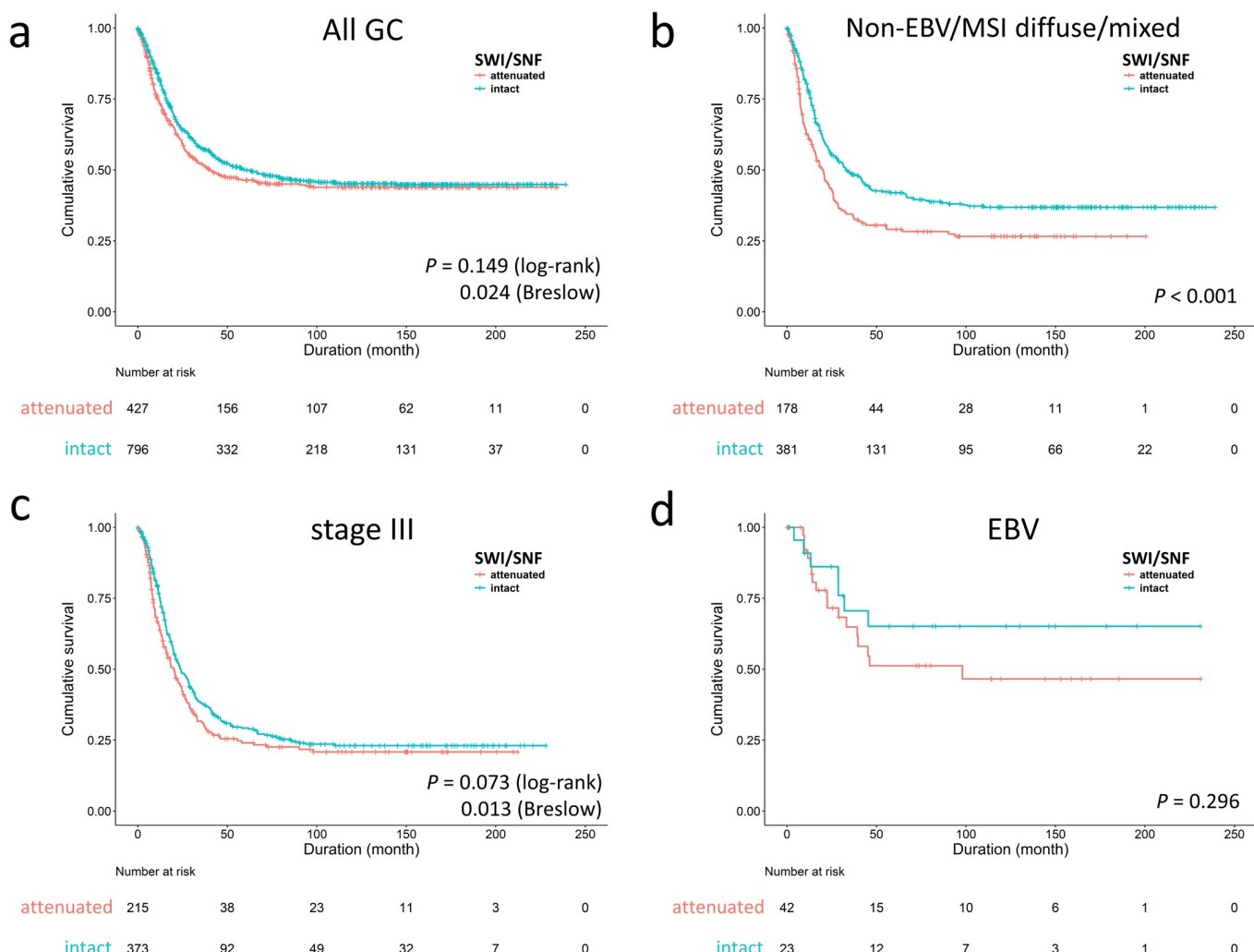

**Fig 2. Survival analysis of SWI/SNF-attenuated gastric cancer (GC). a.** The inferior outcome of SWI/SNF-attenuated GCs occurred in the early time period (*P* = 0.024 in Breslow test). The survival impact of SWI/SNF-attenuated GC was significant in the non-EBV/MSI diffuse/mixed subtype **(b)** and stage III disease **(c)**. **d.** In EBV-associated GC, attenuated SWI/SNF expression had a trend toward poor prognosis (*P* = 0.296).

reduced HER2 positivity were noted in SWI/SNF-attenuated GC (5% versus 8%, *P* = 0.079). In the multivariate logistic regression model, SWI/SNF status was significantly associated with EBV and MSI genotypes [*P* < 0.001; EBV, odds ratio (OR) 3.995, 95% confidence interval (CI) 2.228–7.164; MSI, OR 2.593, 95% CI 1.626–4.136 in reference to the EBV/MSI-negative diffuse/mixed subtype].

For overall survival, attenuated SWI/SNF expression was an independent factor for unfavorable outcome [*P* = 0.001, hazard ratio (HR) 1.360, 95% CI 1.138–1.625] (S1 Table). The univariate log-rank or Breslow analysis showed inconsistent results (*P* = 0.149 and 0.024, respectively), denoting the survival discrimination in SWI/SNF status occurring at early time periods (<5 years) (Fig 2A). In stratification by our proposed genotypes and AJCC stages, the prognostic effect of SWI/SNF status was derived from the EBV/MSI-negative diffuse/mixed subtype (*P* < 0.001, median survival 18.2 versus 31.6 months, Fig 2B) and stage III (*P* = 0.073, 20 versus 24 months; Fig 2C). In EBV-associated GC, cases with attenuated SWI/SNF status had a trend toward unfavorable prognosis (*P* = 0.296) (Fig 2D). For adjusting the influence of chemotherapy, we further evaluated the prognostic importance of SWI/SNF status in 949 cases

of stage II-IV disease (S2 Table). In 946 cases with available information, 703 cases (74%) received chemotherapy and 243 cases (26%) did not. The chemotherapeutic agents were routinely administered postoperatively until patients declined or had contraindications. Since this cohort was retrospectively collected from earlier time, the chemotherapeutic regimens were inconsistent and, in 682 cases with available data, most patients (678, 99%) received 5-fluorouracil-based regimens in the form of single agent (415, 61%, oral or intravenous) or various combinations (263, 39%). The multivariate regression model identified SWI/SNF status was an independent unfavorable parameter ($P$ = 0.019, HR 1.291, 95% CI 1.043–1.597).

## GC with attenuated SMARCA2, SMARCA4, SMARCB1 or ARID1A expression

For better understanding of the clinicopathologic significance of individually altered SWI/SNF subunits, we examined GC with attenuated SMARCA2 and ARID1A expression according to expression patterns. Being the largest population of SWI/SNF-attenuated GC, patients with SMARCA2-attenuated GC also received more total gastrectomy ($P$ = 0.021, S3 Table). There were several clinicopathological variations among SMARCA2-lost, -reduced, -heterogeneous and -retained GCs. SMARCA2-lost/reduced GCs occurred more frequently in the EBV genotype ($P$ < 0.001) and had higher lymph node ratios (ratio of metastatic to retrieved lymph nodes, $P$ = 0.034) and lymphatic invasion ($P$ < 0.001) with a tendency toward pN3 category ($P$ = 0.099). The SMARCA2-lost subgroup demonstrated a higher proportion in poor differentiation, Lauren diffuse/mixed histotype and deeper invasion (pT4 category) ($P$ = 0.005, 0.031 and 0.062, respectively). The SMARCA2-attenuated GC, representing the majority of SWI/SNF-altered GC, nearly recapitulated the prognostic effects of SWI/SNF-attenuated GCs. SMARCA2 attenuation, regardless of loss, reduced and heterogeneous pattern, was associated with inferior overall survival in the early disease time ($P$ = 0.003, Breslow test, Fig 3A), which were derived from the non-EBV/MSI diffuse/mixed subtype ($P$ < 0.001, Fig 3B) and stage III ($P$ = 0.003, Breslow test, Fig 3C). Using backward elimination, the multivariate Cox regression model identified SMARCA2 attenuation as an independent prognostic factor ($P$ = 0.018, HR 1.301, 95% CI 1.046–1.620). For stage II-IV cases with consideration of chemotherapy, SMARCA2 attenuation remained an unfavorable indicator for overall survival ($P$ = 0.018, HR 1.312, 95% CI 1.048–1.643). However, the ratio of patients receiving chemotherapy were different in the SMARCA2-attenuated and SMARCA2-retained groups (85/277 = 31% versus 158/669 = 24%, $P$ = 0.024), indicating imbalance existed in the receipt of adjuvant chemotherapy. Therefore, we did the subgroup analysis and found SMARCA2 attenuation was an unfavorable factor in patients not receiving chemotherapy rather than in those receiving chemotherapy (log-rank test, $P$ = 0.032 and 0.447, respectively). In stage II-IV cases with receiving adjuvant chemotherapy (n = 703), the multivariate Cox regression analysis using backward elimination demonstrated SMARCA2 attenuation was not an independent unfavorable parameter for overall survival ($P$ = 0.204, HR = 1.185, 95% CI 0.912–1.539). The facts suggested patients with SMARCA2 attenuated-GC might benefit from 5-fluorouracil-based chemotherapy.

In contrast, ARID1A-attenuated GCs existed more frequently in both the EBV and MSI genotypes (EBV and MSI cases in ARID1A-attenuated or retained GC, 44% versus 10%, $P$ < 0.001, S4 Table). Attenuated ARID1A expression did not exert a significant impact on overall survival ($P$ = 0.458, Fig 3D) and was not an independent prognostic factor ($P$ = 0.990, HR 0.999, 95% CI 0.877–1.139). However, we observed ARID1A-lost/heterogeneous GCs exhibited a more aggressive behavior in the EBV genotype ($P$ = 0.016, Fig 3E). For SMARCB1-attenuated GCs, only 2 of 11 cases were undifferentiated carcinoma. SMARCB1

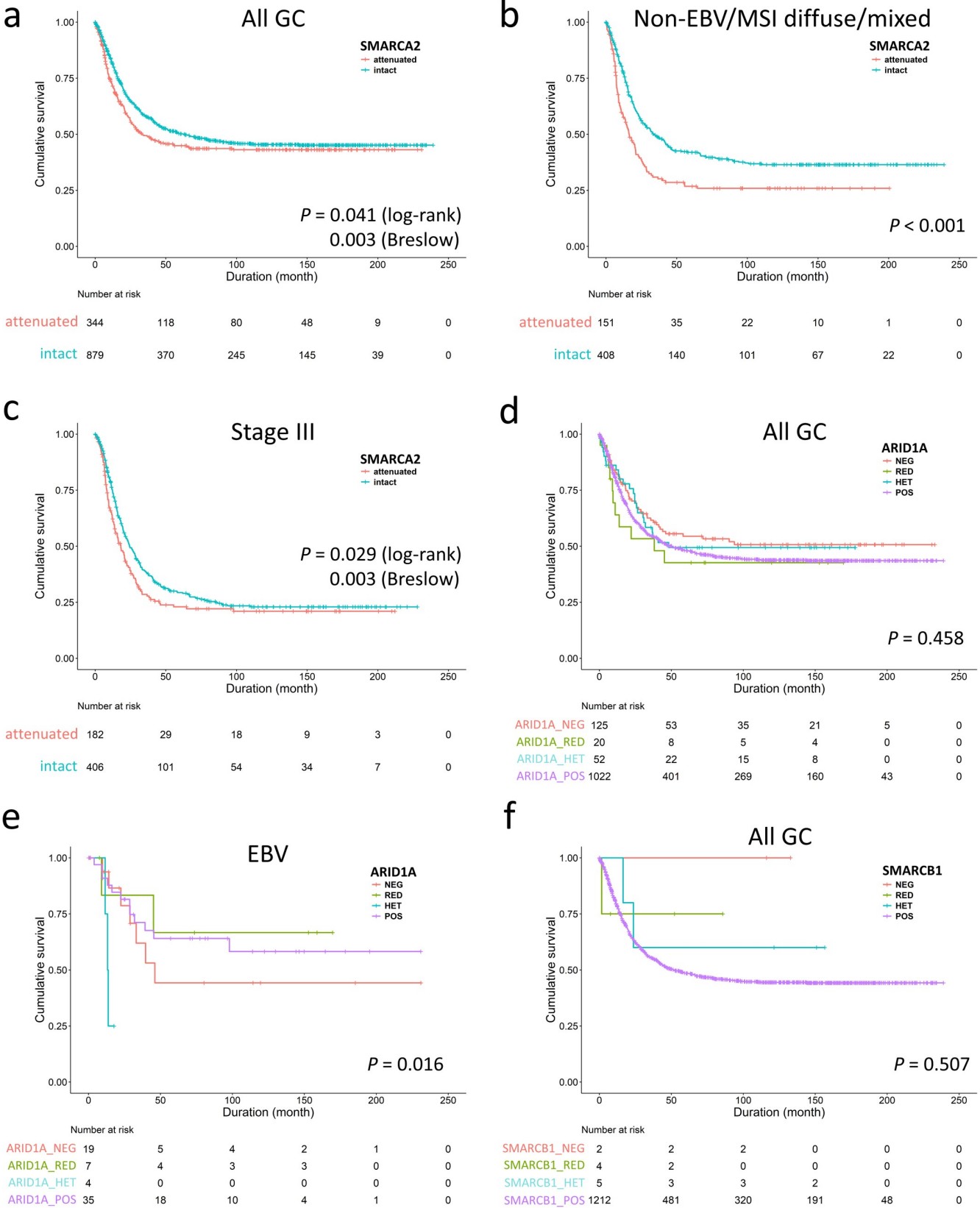

**Fig 3.** The unfavorable survival of SMARCA2-attenuated gastric cancer (GC) was observed in the early disease course **(a)**, the non-EBV/MSI diffuse/mixed subtype **(b)** and stage III disease **(c),** indicating SMARCA2 alteration is the major prognostic effect in SWI/SNF-attenuated GC. **d.** The entire ARID1A-attenuated group had no prognostic significance compared to ARID1A-retained cases. **e.** Subgroup analysis identified ARID1A-lost/heterogeneous expression was associated with unfavorable outcome in only EBV-associated gastric cancer. **f.** SMARCB1-attenuated gastric cancer exhibited no prognostic significance.

attenuation had no survival impact ($P$ = 0.507, Fig 3F) although case numbers were limited. The result of SMARCA4 expression in GC was reported in our previous study [7].

## SWI/SNF-mutated GCs in the TCGA cohort

Since the SWI/SNF complex is considered a tumor suppressor, we restricted GCs with deleterious mutations as SWI/SNF-mutated GCs, including homozygous deletions, insertions/deletions, nonsense/frameshift/splice-site mutations and missense mutations of pathogenetic significance. Missense mutations were determined to be deleterious if PanSoftware or any two of CHASM, CTAT-cancer, DEOGEN2, and PrimateAI algorithms identified the effects of amino acid changes as damaging [19, 20]. In total, 223 of 434 STAD samples (52%) harbored pathogenetic alterations in at least one SWI/SNF subunit, including *ARID1A* (118, 27%), *SMARCA2* (40, 9%), *ARID2* (38, 9%), *ARID1B* (34, 8%), *PBRM1* (32, 7%), *SMARCA4* (29, 7%) and *BCL11A* (25, 6%) (Fig 4A). *ARID2, SMARCA2, ARID1B, PBRM1, SMARCA4* and *BCL11A* mutations significantly coexisted with *ARID1A* mutations ($P$ < 0.05). SWI/SNF mutations more frequently occurred in EBV, MSI and *POLE*-inactivated GCs compared to the GS and CIN subtypes (73%, 97%, 86%, 34%, 38%, respectively, $P$ < 0.001). The prognosis of patients with SWI/SNF-mutated GCs was better than for patients with wild type SWI/SNF GCs in disease-specific survival ($P$ = 0.013) but not in disease-free or progression-free survival ($P$ = 0.858 and 0.269, respectively; Fig 4B). Multivariate Cox regression revealed that SWI/SNF mutation was not an independent prognostic factor ($P$ = 0.550, HR 0.859, 95% CI 0.523–1.413) in consideration of AJCC staging and GC genotypes.

## Discussion

In this study, we performed SMARCA2, SMARCA4, SMARCB1 and ARID1A IHC, a protein-based approach, to identify SWI/SNF-attenuated GC and to investigate molecular alterations using the TCGA STAD dataset. We divided attenuated SWI/SNF expression into 3 patterns according to up-to-date classification schemes [16, 17]. These 3 patterns are all regarded as SWI/SNF-attenuated GC based on previous observations that the SWI/SNF-lost phenomenon is caused by molecular alterations in the corresponding SWI/SNF subunits *per se*, and the SWI/SNF-reduced pattern is linked to secondary diminishment from alterations in other SWI/SNF subunits [7, 21]. The concurrent attenuation of multiple SWI/SNF subunits was identified in 31% of cases, illustrating the intimate interaction among SWI/SNF subunits.

In our study, altered SMARCA2 protein expression was the most common phenomenon (28%) and was associated with unfavorable prognosis in the non-EBV/MSI diffuse/mixed subgroup and stage III disease. Low SMARCA2 expression in GC has been previously described by Yamamichi and colleagues [8]. They reported that SMARCA2 expression was severely decreased (>50% of tumor cells were negative) in 42% (37/89) of GCs and deficient SMARCA2 expression was usually in tubular and papillary adenocarcinoma but not in signet-ring cell or mucinous carcinoma. By current molecular subtyping, we identified SMARCA2-lost/reduced GCs occurring more frequently in the EBV genotype (17% and 12%, respectively, versus 3% in the SMARCA2-retained GCs, $P$ < 0.001), and the majority of SMARCA2-lost GCs exhibited poor differentiation and Lauren diffuse/mixed histologic features (76% and

a

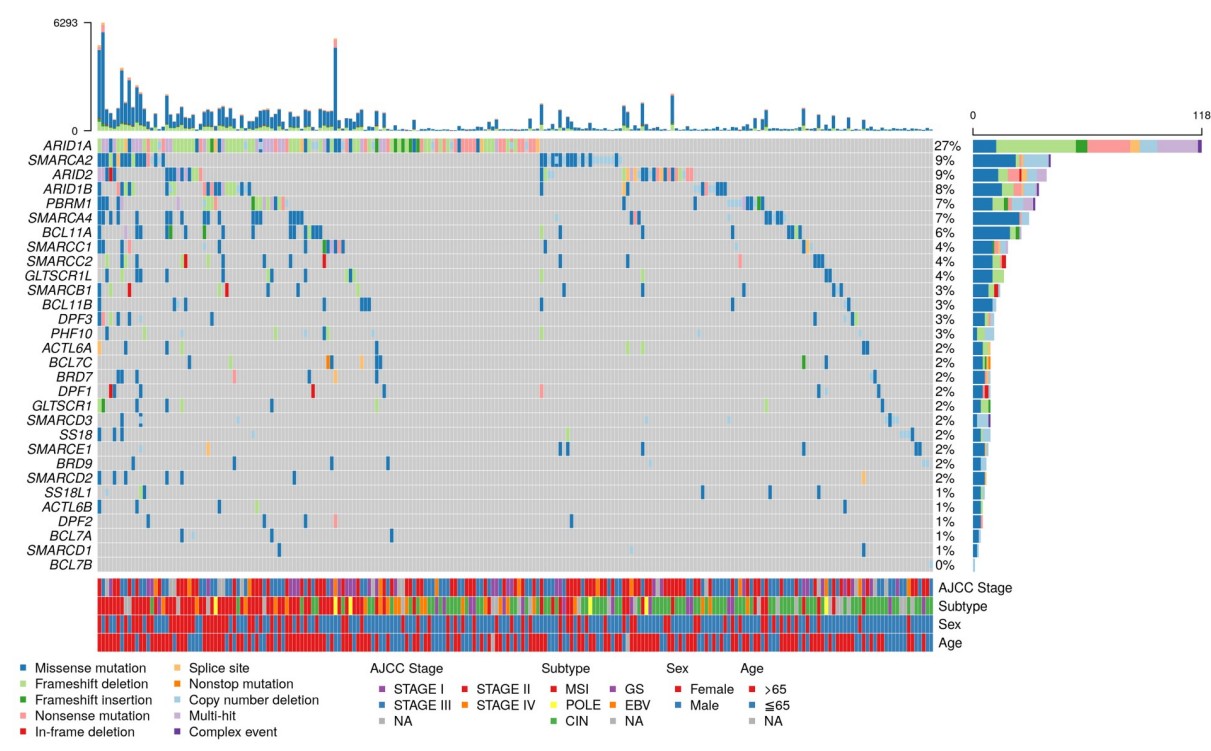

b

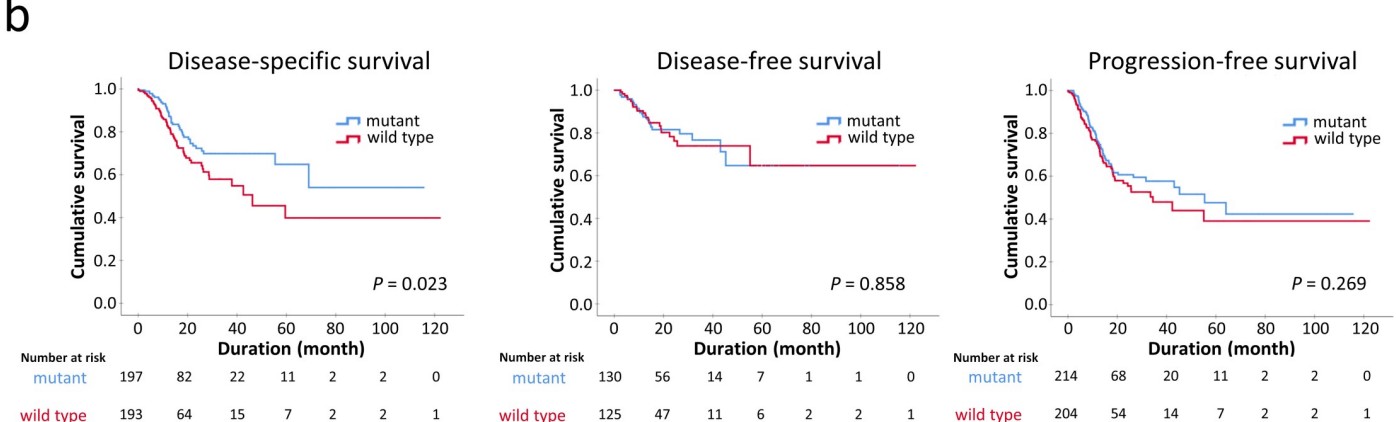

**Fig 4. a.** The oncoplot showing the landscape of SWI/SNF mutations in 434 STAD patients from the TCGA Pan-Cancer Atlas. Synchronous multiple SWI/SNF mutations were common, especially in the microsatellite-instable genotype. The upper part represents the mutation burden of each patient. NA, not available. **b.** SWI/SNF-mutated gastric cancer exhibits an association with improved disease-specific survival ($P = 0.023$) but had no significant prognostic difference in disease-free or progression-free survival.

63%, respectively, $P < 0.05$). Due to the histologic and prognostic significance, we suppose that SMARCA2-lost GC might represent a distinct molecular subgroup of non-EBV/MSI diffuse/mixed GC that deserves SMARCA2-targeted therapy.

Decreased ARID1A expression was the second most common event (16%) of SWI/SNF defect in our cohort. The incidence of ARID1A loss in GC ranged from 8% to 70% (median 25%) using various cutoff levels defined as cancer cells weak or without nuclear staining, or nuclear staining < 10% [6]. In agreement with previous reports [22], we found GCs with

ARID1A attenuation were significantly associated with EBV and MSI status compared to ARID1A-retained GC (16% and 28% versus 4% and 6% for EBV and MSI, respectively, $P < 0.001$). ARID1A-attenuated GC has no characteristic clinicopathologic features, except for a predilection to stump cancer and low HER2 positivity ($P < 0.05$). The former might be due to enriched EBV-positive cases in ARID1A-attenuated GC. Although a meta-analysis identified ARID1A loss as associated with poor overall survival (HR 1.60, 95% CI 1.40–1.81, $P < 0.001$) [6], we were unable to confirm the prognostic effect of ARID1A defects in our cohort ($P = 0.990$, HR 0.999, 95% CI 0.877–1.139). Nevertheless, the subgroup analysis demonstrated that EBV-associated GC with ARID1A-lost/heterogeneous expression exhibited more aggressive behavior ($P = 0.016$), corroborated by the meta-analysis revealing that ARID1A loss was associated with poor overall survival in EBV-associated GC > 5% subgroup (HR 1.59, 95% CI 1.18–2.15) [6]. EBV might play synergistically with ARID1A alteration in progression of GC.

Emerging data indicates that SWI/SNF alterations result in vulnerabilities in cancers, through directly targeting SWI/SNF complexes, targeting PRC2 via EZH2, or targeting downstream deregulation [9, 10]. In addition, SWI/SNF-altered cancers are also sensitive to DNA damage repair and immune-checkpoint inhibitors [10]. The ongoing clinical trials have been tested several therapeutic agents in patients whose cancers harbor SWI/SNF aberrations. An EZH2 inhibitor, tazemetostat, just gained approval for treatment of epithelioid sarcoma harboring SMARCB1 loss in January 2020 in the USA [23]. Our current study demonstrated SMARCA2-attenuated GC exhibited more aggressive course in AJCC stage 3 and non-EBV/MSI diffuse/mixed subtype and ARID1A alteration was associated with more inferior survival in patients with EBV-associated GC. These findings not only suggest SMARCA2 alteration might supplement the TNM stage in clinical settings but also implicate the SMARCA2 or ARID1A-targeted management could impart more benefit in non-EBV/MSI diffuse/mixed or EBV-associated GCs, respectively. The probable explanation is that the biologic effect of SWI/SNF impairment is associated with the accompanying genetic context as a result of SWI/SNF complex serving a chromatin remodeler controlling global transcription [10]. This information offers the possible directions of further research in SWI/SNF-altered GC.

SMARCB1 deficiency was linked to malignant rhabdoid tumors and the literature reported gastric rhabdoid/undifferentiated carcinomas were associated with complete SMARCB1 absence [24, 25]. Our present study found that SMARCB1-attenuated GCs are very rare (11/1224, 1%), with only 2 genuine SMARCB1-lost cases and the remaining cases being either reduced or heterogeneous expression. Except for 2 cases with undifferentiated carcinoma, SMARCB1-attenuated GC could be a tubular or poorly cohesive carcinoma. The above result is similar to our previous finding that SMARCA4-lost GC appeared not only in undifferentiated/rhabdoid carcinoma but also in tubular adenocarcinoma [7]. For lung cancer, complete SMARCA4 loss indeed existed in 5% of adenocarcinomas and squamous cell carcinomas [17]. In a genetically engineered mouse model, sole *Smarca4* knockout failed to induce lung adenocarcinoma, while concurrent introduction of *p53* inactivation and *Kras* mutations resulted in robust development of highly penetrant undifferentiated carcinomas, indicating the requirement of additional genetic alterations in SMARCA4-deficit tumors to drive undifferentiated progression [26].

Furthermore, we used the STAD dataset of the TCGA Pan-Cancer Atlas and found 223 (52%) of 434 samples harbored deleterious SWI/SNF mutations. Concurrent multiple mutations were observed in one-third of cases, especially for *ARID1A* mutations and in MSI-related GC. Intriguingly, SWI/SNF-mutated GC demonstrated favorable disease-specific survival, which might be attributed to the enriched MSI genotype in SWI/SNF-mutated GCs (35% versus 1% in SWI/SNF-mutated and wild type GCs, respectively). As MSI causes highly mutated

genomic profiling, the frequent SWI/SNF mutations in MSI-associated GC may be a second phenomenon, instead of primary event for target therapies.

The TCGA STAD data showed some discrepancies between our study. The first one is the frequencies of SMARCA2 attenuation and mutation. SMARCA2 attenuation is the largest sub-group in SWI/SNF-attenuated GCs (27%), but deleterious *SMARCA2* mutations exist in only 9% of GCs in TCGA data. Previous studies have shown that most SMARCA2 inactivation is driven by epigenetic silencing rather than abrogating mutations [8, 27]. This fact exemplifies the importance of applying IHC to detect SMARCA2-attenuated GC. The other limitation of our study is lacking data for ARID2, PBRM1, GLTSCR1, GLTSCR1L, etc., which are specific subunits for PBAF (Polybromo-associated BRG1/BRM-associated factor; BAF = mammalian SWI/SNF) and ncBAF (non-canonical BAF), respectively. These newly discovered SWI/SNF subunits might have biologic significance. ncBAF has been described as a synthetic lethal target in cancers driven by deficient BAF complex [28]. *PBRM1* mutations are linked to immuno-therapy response in patients with metastatic renal cell carcinoma [29]. Additional studies are needed to clarify the significance of PBAF and ncBAF in GC.

## Conclusions

In conclusion, we examined SMARCA2, SMARCA4, SMARCB1 and ARID1A attenuation and SWI/SNF mutations in GC and observed that clinical significance was primarily related to genotype. Both SWI/SNF attenuation and mutations were more prevalent in EBV and/or MSI subgroups. SMARCA2 and ARID1A attenuation has unfavorable effects in non-EBV/MSI dif-fuse/mixed and EBV subtypes, respectively. The SWI/SNF mutations are enriched in MSI genotype, possibly due to hypermutated profiling. As our knowledge of the SWI/SNF complex continues to grow, more studies are needed to reveal the biologic consequence and clinical sig-nificance of SWI/SNF perturbations incorporating the knowledge of GC molecular subtypes.

## Supporting information

**S1 Table. Univariate and multivariate analysis of prognostic factors in patients with gastric cancer according to overall survival.**
(DOC)

**S2 Table. Univariate and multivariate analysis of prognostic factors in patients with stage II-IV gastric cancer, including chemotherapy data.**
(DOC)

**S3 Table. Clinicopathological differences between SMARCA2-retained and SMARCA2-at-tenuated gastric cancers.**
(DOCX)

**S4 Table. Clinicopathological differences between ARID1A-retained and ARID1A-attenu-ated gastric cancers.**
(DOCX)

## Acknowledgments

The TCGA data analysis was performed by the Bioinformatics Core Laboratory, Molecular Medicine Research Center, Chang Gung University, Taiwan, supported by the Chang Gung Memorial Hospital (CLRPD1J0012) and the "Molecular Medicine Research Center, Chang Gung University" from The Featured Areas Research Center Program within the framework of the Higher Education Sprout Project by the Ministry of Education in Taiwan.

## Author Contributions

**Conceptualization:** Shih-Chiang Huang, Tse-Ching Chen.

**Data curation:** Shih-Chiang Huang, Ta-Sen Yeh.

**Formal analysis:** Shih-Chiang Huang, Ian Yi-Feng Chang, Chee-Jen Chang, Shu-Chen Chang, Min-Chi Chen.

**Funding acquisition:** Shih-Chiang Huang, Tse-Ching Chen.

**Investigation:** Shih-Chiang Huang, Kwai-Fong Ng, Yi-Chun Chao.

**Methodology:** Shih-Chiang Huang, Ian Yi-Feng Chang, Yi-Chun Chao.

**Project administration:** Tse-Ching Chen.

**Resources:** Ta-Sen Yeh, Tse-Ching Chen.

**Software:** Ian Yi-Feng Chang.

**Visualization:** Ian Yi-Feng Chang.

**Writing – original draft:** Shih-Chiang Huang.

**Writing – review & editing:** Tse-Ching Chen.

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
