## [Decision Letter · Decision Letter 0]

4 Nov 2020

PONE-D-20-31951

The clinicopathological significance of SWI/SNF alterations in gastric cancer is associated with the molecular subtypes

PLOS ONE

Dear Dr. Chen,

Thank you for submitting your manuscript to PLOS ONE. After careful consideration, we feel that it has merit but does not fully meet PLOS ONE’s publication criteria as it currently stands. Therefore, we invite you to submit a revised version of the manuscript that addresses the points raised during the review process.

This study was carefully reviewed by 2 experts, and both of them found a number of concerns and questions which need to be addressed before this manuscript becomes potentially acceptable. For instance, reviewer 1 suggested validation using independent dataset. Also, reviewer 2 suggested a potential problem in the classification used in this study. Please respond to each of the reviewer comments.

We look forward to receiving your revised manuscript.

Kind regards,

Hiromu Suzuki, M.D., Ph.D.

Academic Editor

PLOS ONE

Journal Requirements:

2. In the ethics statement in the manuscript and in the online submission form, please provide additional information about the patient records/samples used in your retrospective study, including: a) whether all data were fully anonymized before you accessed them; b) the date range (month and year) during which patients' medical records/samples were accessed.

3. Please provide accession numbers and/or URLs for the TCGA dataset analysed.

4. At this time, we ask that you please provide scale bars on the microscopy images presented in Figure 1and refer to the scale bar in the corresponding Figure legend.

5.  Thank you for stating in your Funding Statement:

"This work was supported by grants from the Ministry of Science and Technology (108-2320-B-182A-018, 106-2320-B-182A-011-MY3 and 105-2320-B-182A-014) and the Chang Gung Memorial Hospital (CMRPG3F2073, CMRPG5K0021, CIRPG3D0153, CMRP3C1323 and CMRPG3G0553).".

i) Please provide an amended statement that declares *all* the funding or sources of support (whether external or internal to your organization) received during this study, as detailed online in our guide for authors at http://journals.plos.org/plosone/s/submit-now.  Please also include the statement “There was no additional external funding received for this study.” in your updated Funding Statement.

ii) Please include your amended Funding Statement within your cover letter. We will change the online submission form on your behalf.

6.  We note that you have indicated that data from this study are available upon request. PLOS only allows data to be available upon request if there are legal or ethical restrictions on sharing data publicly. For information on unacceptable data access restrictions, please see http://journals.plos.org/plosone/s/data-availability#loc-unacceptable-data-access-restrictions.

Reviewers' comments:

Reviewer's Responses to Questions

**Comments to the Author**

1. Is the manuscript technically sound, and do the data support the conclusions?

Reviewer #1: Partly

Reviewer #2: Yes

2. Has the statistical analysis been performed appropriately and rigorously? 

Reviewer #1: No

Reviewer #2: Yes

3. Have the authors made all data underlying the findings in their manuscript fully available?

Reviewer #1: No

Reviewer #2: Yes

4. Is the manuscript presented in an intelligible fashion and written in standard English?

Reviewer #1: Yes

Reviewer #2: Yes

5. Review Comments to the Author

Reviewer #1: Summary: Huang et al reported that molecular-level alterations of SWI/SNF components complex including SMARCA2, SMARCA4, SMARCCB1, and ARID1A are associated with clinicopathological findings of gastric cancer. The retrospective cohort includes >1,000 surgically-resected samples. The genetic analyses were of pre-published data and the overall prognostic impact of any of presenting new parameters seems marginal. Therefore, despite of the good size of cohort, overall conclusion is not striking.

1. This is a knowledge-oriented survey, not a data-driven research. It is eventually required an independent study (i.e., prospective) to prove the present clinicopathological significance. However, it would be very difficult to conduct if the motive is from merely a literature search. It would be nice to add what authors expect by publishing this study.

2. An association is not a significance. For instance, which new parameters have been suggested to be better predictive markers than TNM staging or ly or v pathologic parameters? Who will need the new data?

3. What is eligible criteria for chemotherapy? In general, post-operative adjuvant chemotherapy is the most responsible prognostic marker for Stage II/III GCs. The demand of GC research is to discover stratifying markers, such as predicting good prognosis from stage IV patients.

4. The definition of subgroup analysis was not clear throughout the manuscript. Reviewer suggests to use a proportional hazards analysis along with interaction p value instead of a group of survival curves.

5. Numbers at risk must be added to each of survival curves.

6. The resolution of figure legends was unreadable.

Reviewer #2: In this study, the authors evaluated the expression of four SWI/SNF subunits by immunohistochemistry and analyzed SWI/SNF mutation using TCGA data in gastric cancer. They showed SWI/SNF attenuations and mutations were associated with EBV and MSI molecular subtypes of gastric cancer.

Minor comments:

1. (Table 1) Intestinal and Diffuse/Mixed types are Lauren classification; therefore, Genotypes should be divided into EBV, MSI, and Other (non-EBV/MSI), rather than EBV, MSI, Intestinal, and Diffuse/mixed.

2. (Method and Table 1) There is no detail information about chemotherapy, do you mean chemotherapy is adjuvant chemotherapy? Patients with neoadjuvant chemotherapy were excluded from these cohorts?

3. (16p, "For stage II-IV cases with consideration of chemotherapy, SMARCA2 attenuation remained an unfavorable indicator for overall survival (P = 0.001, HR 1.388, 95% CI 1.144-1.685), suggesting SMARCA2-altered GC might retain the intrinsic property of chemoresistance.")

: Did all stage II-IV patients received adjuvant chemotherapy? If not, the authors should have shown that the results of stage II-IV cases with adjuvant chemotherapy. In 1999-2007, the adjuvant chemotherapy regimen might be varied. Do you have any data on what adjuvant chemotherapy regimen of patients have treated? Although unlikely, an imbalance in the receipt of adjuvant chemotherapy in these SMARCA2-attenuated vs. SMARCA2-retained groups could be a confounding variable.

4. In Figure 3a-c figure, it would be better to display a comparison of the groups by dividing into two groups (SMARCA2-attenuated and SMARCA2-retained), rather than comparing the groups by dividing them into four groups (HET, NEG, POS, and RED).

5. (Figure2-3) Although Figure and Legend are understandable with the careful reading of the text, it is not well presented so that the result is not easily recognized. The following is not a request but a suggestion.

In figure3, it is better to use “all GC”, “non-EBV/MSI diffuse/mixed”, “all GC” and “all GC” instead of “SMARCA2”, “diffuse/mixed” “ARID1A” and “SMARCB1”. And, it is better to use “SMARCA2_HET”, “SMARCA2_NEG”, “SMARCA2_POG”, and “SMARCA2_RED” instead of “HET”, “NEG”, “POG” and “RED”.

In figure3e legend, it is better to use “ARID1A-lost/heterogeneous” instead of “attenuated ARID1A expression”

6. (15p) It is better to use “pT4 category” and “pN3 category” instead of “T4 stage” and “N3 stage”

6. PLOS authors have the option to publish the peer review history of their article (what does this mean?). If published, this will include your full peer review and any attached files.

Reviewer #1: No

Reviewer #2: No

---

## [Author Response · Author response to Decision Letter 0]

12 Dec 2020

Dr. Hiromu Suzuki 

Academic Editor 

Editorial Office, PLOS ONE

December 7th, 2020

Dear Dr. Hiromu Suzuki, 

Thanks for your valuable comments on our manuscript entitled “The clinicopathological significance of SWI/SNF alterations in gastric cancer is associated with the molecular subtypes” (Manuscript ID: PONE-D-20-31951). As requested by reviewers, we have revised our manuscript according to their suggestions point by point, highlighted the pertinent changes in red, and briefly summarized these revisions as follows. We highly appreciate this opportunity of improving the quality of our article through addressing all the comments raised by reviewers and the manuscript is re-submitted for your consideration. 

Funding statement

This work was supported by grants from the Ministry of Science and Technology (108-2320-B-182A-018, 106-2320-B-182A-011-MY3 and 105-2320-B-182A-014) and the Chang Gung Memorial Hospital (CMRPG3F2073, CMRPG5K0021, CIRPG3D0153, CMRP3C1323 and CMRPG3G0553). The funders had no role in study design, data collection and analysis, decision to publish, or preparation of the manuscript. All the funding or sources of support were received during this study and there was no additional external funding received for this study.

Availability of data and materials 

Data cannot be shared publicly due to potentially identifying and sensitive patient information. Data are only available with approval of the Institutional Ethics Committee of the Chang Gung Memorial Hospital (irb1@cgmh.org.tw) or via the corresponding author (ctc323@cgmh.org.tw) for researchers who meet the criteria for access to confidential data.

Sincerely,

Tse-Ching Chen, M.D., Ph.D.

Department of Anatomic Pathology, Linkou Chang Gung Memorial Hospital, No.5, Fuxing St., Guishan Dist., Taoyuan City 33305, Taiwan

E-mail: ctc323@adm.cgmh.org.tw

REPLY TO REVIEWERS' COMMENTS:

Reviewer #1: Summary: Huang et al reported that molecular-level alterations of SWI/SNF components complex including SMARCA2, SMARCA4, SMARCCB1, and ARID1A are associated with clinicopathological findings of gastric cancer. The retrospective cohort includes >1,000 surgically-resected samples. The genetic analyses were of pre-published data and the overall prognostic impact of any of presenting new parameters seems marginal. Therefore, despite of the good size of cohort, overall conclusion is not striking.

1. This is a knowledge-oriented survey, not a data-driven research. It is eventually required an independent study (i.e., prospective) to prove the present clinicopathological significance. However, it would be very difficult to conduct if the motive is from merely a literature search. It would be nice to add what authors expect by publishing this study.

Reply: It is indeed difficult for us to perform a prospective independent study to validate the clinicopathological significance and molecular results. By publishing this study, we would like to disclose the clinical significance of SWI/SNF alterations in gastric cancer. SWI/SNF components have been considered to be therapeutic targets and the application of SWI/SNF-targeted therapy could be beneficial for patients with SWI/SNF-altered gastric cancer. Therefore, selection of appropriate candidates for targeted therapy would become a pivotal issue in the clinical management. We found it could be more precise in the viewpoint of gastric cancer molecular subtype. The probable explanation is that the biologic effect of SWI/SNF impairment is associated with the accompanying genetic context as a result of SWI/SNF complex serving a chromatin remodeler controlling global transcription.

Manuscript, page 23, line 372-389: Emerging data indicates that SWI/SNF alterations result in vulnerabilities in cancers, through directly targeting SWI/SNF complexes, targeting PRC2 via EZH2, or targeting downstream deregulation [9, 10]. In addition, SWI/SNF-altered cancers are also sensitive to DNA damage repair and immune-checkpoint inhibitors [10]. The ongoing clinical trials have been tested several therapeutic agents in patients whose cancers harbor SWI/SNF aberrations. An EZH2 inhibitor, tazemetostat, just gained approval for treatment of epithelioid sarcoma harboring SMARCB1 loss in January 2020 in the USA [23]. Our current study demonstrated SMARCA2-attenuated GC exhibited more aggressive course in AJCC stage 3 and non-EBV/MSI diffuse/mixed subtype and ARID1A alteration was associated with more inferior survival in patients with EBV-associated GC. These findings not only suggest SMARCA2 alteration might supplement the TNM stage in clinical settings but also implicate the SMARCA2 or ARID1A-targeted management could impart more benefit in non-EBV/MSI diffuse/mixed or EBV-associated GCs, respectively. The probable explanation is that the biologic effect of SWI/SNF impairment is associated with the accompanying genetic context as a result of SWI/SNF complex serving a chromatin remodeler controlling global transcription [10]. This information offers the possible directions of further research in SWI/SNF-altered GC.

Reference:

[9] St Pierre R, Kadoch C. Mammalian SWI/SNF complexes in cancer: emerging therapeutic opportunities. Curr Opin Genet Dev. 2017;42:56-67.

[10] Mittal P, Roberts CWM. The SWI/SNF complex in cancer - biology, biomarkers and therapy. Nat Rev Clin Oncol. 2020;17:435-448.

[23] Hoy SM. Tazemetostat: First Approval. Drugs. 2020;80(5):513-521.

2. An association is not a significance. For instance, which new parameters have been suggested to be better predictive markers than TNM staging or ly or v pathologic parameters? Who will need the new data?

Reply: In our current study, the AJCC stages and GC genotypes, along with gastrectomy type, extent of lymph node dissection, and usage of chemotherapy, remained the most critical prognostic parameters in our analysis (S1 and S2 Table). We discovered the SWI/SNF alteration, i.e. SMARCA2 attenuation, could improve the predicting prognosis in stage III and the prognostic significance of SMARCA2 and ARID1A alterations had association with GC subtypes. This information might supplement the TNM stage in clinical settings and offer the possible directions of further research. 

Manuscript, page 23, line 382-389: These findings not only suggest SMARCA2 alteration might supplement the TNM stage in clinical settings but also implicate the SMARCA2 or ARID1A-targeted management could impart more benefit in non-EBV/MSI diffuse/mixed or EBV-associated GCs, respectively. The probable explanation is that the biologic effect of SWI/SNF impairment is associated with the accompanying genetic context as a result of SWI/SNF complex serving a chromatin remodeler controlling global transcription [10]. This information offers the possible directions of further research in SWI/SNF-altered GC.

Reference:

[10] Mittal P, Roberts CWM. The SWI/SNF complex in cancer - biology, biomarkers and therapy. Nat Rev Clin Oncol. 2020;17:435-448.

3. What is eligible criteria for chemotherapy? In general, post-operative adjuvant chemotherapy is the most responsible prognostic marker for Stage II/III GCs. The demand of GC research is to discover stratifying markers, such as predicting good prognosis from stage IV patients.

Reply: The chemotherapeutic agents were routinely administrated postoperatively until patients declined or had contraindications. In this study, we indeed identified the adjuvant chemotherapy was an independent prognostic factor for patients with stage II-IV disease (S2 Table; P < 0.001, HR 0.595, 95% CI 0.462-0.766) and SWI/SNF alteration was also a useful stratifying biomarker (S2 Table; P = 0.019, HR 1.291, 95% CI 1.043-1.597). The further analysis demonstrated the major component of SWI/SNF alteration, i.e. SMARCA2 attenuation, was an independent prognostic biomarker (P = 0.018, HR 1.312, 95% CI 1.048-1.643).

Manuscript, page 15-16, line 225-235: For adjusting the influence of chemotherapy, we further evaluated the prognostic importance of SWI/SNF status in 949 cases of stage II-IV disease (S2 Table). In 946 cases with available information, 703 cases (74%) received chemotherapy and 243 cases (26%) did not. The chemotherapeutic agents were routinely administrated postoperatively until patients declined or had contraindications. Since this cohort was retrospectively collected from earlier time, the chemotherapeutic regimens were inconsistent and, in 682 cases with available data, most patients (678, 99%) received 5-fluorouracil-based regimens in the form of single agent (415, 61%, oral or intravenous) or various combinations (263, 39%). The multivariate regression model identified SWI/SNF status was an independent unfavorable parameter (P = 0.019, HR 1.291, 95% CI 1.043-1.597).

Manuscript, page 17, line 264-266: For stage II-IV cases with consideration of chemotherapy, SMARCA2 attenuation remained an unfavorable indicator for overall survival (P = 0.018, HR 1.312, 95% CI 1.048-1.643).

4. The definition of subgroup analysis was not clear throughout the manuscript. Reviewer suggests to use a proportional hazards analysis along with interaction p value instead of a group of survival curves.

Reply: For the clinicopathologic factors appearing significant in multivariate Cox regression analysis, we progressed to perform subgroup analysis to determine which subgroup was more susceptible for the alterations of SWI/SNI component. Among the most significant independent factors for survival (P ≤ 0.001; gastrectomy and lymphadenectomy type, combined classification, AJCC stages, chemotherapy treatment), combined classification and AJCC stages likely represented tumor biologic properties and other factors indicated clinical interventions. Therefore, we focused on the subgroup analysis in these two factors. Since the incidence of SWI/SNF alterations was significantly related to the combined classification (P < 0.001, Table 1), the multicollinearity would occur in a regression proportional hazards analysis for interaction p value. This correlation caused a problem because the variables should be independent for the regression proportional hazards analysis. For this sake, we only test the interaction p value for AJCC stage. The interaction analysis result showed significant (P = 0.04), and the subgroup analysis for AJCC stage was done thereafter.

Manuscript, page 9-10, line 154-165: For the clinicopathologic factors appearing significant in multivariate Cox regression analysis, we progressed to perform subgroup analysis to determine which subgroup was more susceptible for the alterations of SWI/SNI component. Among the most significant independent factors for survival (P ≤ 0.001; gastrectomy and lymphadenectomy type, combined classification, AJCC stages, chemotherapy treatment), combined classification and AJCC stages likely represented tumor biologic properties and other factors indicated clinical interventions. The regression proportional hazards analysis for interaction P value was done only for AJCC stage since the incidence of SWI/SNF alterations was significantly related to the combined classification (P < 0.001), leading to the multicollinear problem. The interaction test for AJCC stage showed significant (P = 0.04), and the subgroup analysis was done thereafter.

5. Numbers at risk must be added to each of survival curves.

Reply: We added numbers at risk in each of survival curves.

Manuscript, Fig 2-4.

6. The resolution of figure legends was unreadable. 

Reply: We improved the resolution of figure legends.

Manuscript, Fig 1-4.

Reviewer #2: In this study, the authors evaluated the expression of four SWI/SNF subunits by immunohistochemistry and analyzed SWI/SNF mutation using TCGA data in gastric cancer. They showed SWI/SNF attenuations and mutations were associated with EBV and MSI molecular subtypes of gastric cancer.

Minor comments:

1. (Table 1) Intestinal and Diffuse/Mixed types are Lauren classification; therefore, Genotypes should be divided into EBV, MSI, and Other (non-EBV/MSI), rather than EBV, MSI, Intestinal, and Diffuse/mixed.

Reply: In our previous study [ref. 3], we designed a simple 4-subtype classification (EBV, MSI, and non-EBV/MSI intestinal and diffuse/mixed type) in parallel with the TCGA molecular categorization (EBV, MSI, chromosomal instable, genomically stable). In brief, we first subtracted EBV-positive and MSI-associated GC by EBV-encoded small RNA in situ hybridization (EBER-ISH) and immunohistochemistry of DNA mismatch repair proteins (MMR-IHC), respectively. Afterwards, the remaining GC cases were divided into Lauren intestinal, diffuse, and mixed histotypes. The non-EBV/MSI intestinal subtype is close to the TCGA’s chromosomal instable (CIN) category and the non-EBV/MSI diffuse/mixed subtype to the TCGA’s genomically stable (GS) category. The reasons are most of the CIN (80%) cases are of the intestinal phenotype and diffuse/mixed carcinoma is enriched in the GS (74%) genotype. In our cohort of 1,248 patients with gastric cancer who received radical gastrectomy, the clinical and molecular characteristics of non-EBV/MSI intestinal and diffuse/mixed subtypes are similar to those of CIN and GS categories, respectively. Therefore, the non-EBV/MSI intestinal and diffuse/mixed subtypes in our classification represent distinct TCGA CIN and GS molecular categories, respectively. We propose to preserve this 4-subtype analysis.

Manuscript, page 5, line 70-76: In our previous work, we integrated EBV-encoded small RNA in situ hybridization (EBER-ISH), immunohistochemistry of DNA mismatch proteins (MMR-IHC) and Lauren histotyping to design a practical GC subtyping algorithm, parallel to the TCGA classification [3]. In brief, the Lauren intestinal and diffuse/mixed division was done after EBV and MSI-associated GCs were subtracted. The non-EBV/MSI intestinal and diffuse/mixed subtypes had clinical and molecular similarity to the TCGA CIN and GS variants, respectively [3].

Manuscript, page 8, line 127-131: In this study, GCs were first divided into EBV and MSI, and the remaining negative cases were classified into Lauren intestinal and diffuse/mixed subtypes [3]. According to our previous study [3], the non-EBV/MSI intestinal and diffuse/mixed subtypes approximately represent TCGA’s CIN and GS molecular categories, respectively.

Reference [3]: Huang SC, Ng KF, Yeh TS, Cheng CT, Lin JS, Liu YJ, et al. Subtraction of Epstein-Barr virus and microsatellite instability genotypes from the Lauren histotypes: Combined molecular and histologic subtyping with clinicopathological and prognostic significance validated in a cohort of 1,248 cases. International Journal of Cancer. 2019;145(12):3218-30.

2. (Method and Table 1) There is no detail information about chemotherapy, do you mean chemotherapy is adjuvant chemotherapy? Patients with neoadjuvant chemotherapy were excluded from these cohorts?

Reply: All chemotherapies are adjuvant, i.e. administrated after operation. In 946 cases of stage II-IV with available information, 703 cases (74%) received chemotherapy and 243 cases (26%) did not. The chemotherapeutic agents were routinely administrated postoperatively until patients declined or had contraindications. Since this cohort was retrospectively collected from earlier time, the chemotherapeutic regimens were inconsistent and, in 682 cases with available data, most patients (678, 99%) received 5-fluorouracil-based regimens in the form of single agent (415, 61%, oral or intravenous) or various combinations (263, 39%).

Manuscript, page 15-16, line 225-233: For adjusting the influence of chemotherapy, we further evaluated the prognostic importance of SWI/SNF status in 949 cases of stage II-IV disease (S2 Table). In 946 cases with available information, 703 cases (74%) received chemotherapy and 243 cases (26%) did not. The chemotherapeutic agents were routinely administrated postoperatively until patients declined or had contraindications. Since this cohort was retrospectively collected from earlier time, the chemotherapeutic regimens were inconsistent and, in 682 cases with available data, most patients (678, 99%) received 5-fluorouracil-based regimens in the form of single agent (415, 61%, oral or intravenous) or various combinations (263, 39%). 

3. (16p, "For stage II-IV cases with consideration of chemotherapy, SMARCA2 attenuation remained an unfavorable indicator for overall survival (P = 0.001, HR 1.388, 95% CI 1.144-1.685), suggesting SMARCA2-altered GC might retain the intrinsic property of chemoresistance.") Did all stage II-IV patients received adjuvant chemotherapy? If not, the authors should have shown that the results of stage II-IV cases with adjuvant chemotherapy. In 1999-2007, the adjuvant chemotherapy regimen might be varied. Do you have any data on what adjuvant chemotherapy regimen of patients have treated? Although unlikely, an imbalance in the receipt of adjuvant chemotherapy in these SMARCA2-attenuated vs. SMARCA2-retained groups could be a confounding variable.

Reply: As the reply for comment #2, 946 of 949 cases of stage II-IV disease had available information regarding chemotherapy. There were 703 cases (74%) receiving chemotherapy and 243 cases (26%) not. The chemotherapeutic agents were routinely administrated postoperatively until patients declined or had contraindications. Since this cohort was retrospectively collected from earlier time, the chemotherapeutic regimens were inconsistent and, in 682 cases with available data, most patients (678, 99%) received 5-fluorouracil-based regimens in the form of single agent (415, 61%, oral or intravenous) or various combinations (263, 39%). In the SMARCA2-attenuated and SMARCA2-retained groups, the ratio of patients receiving chemotherapy were different (85/277 = 31% versus 158/669 = 24%), indicating significant imbalance existed in the receipt of adjuvant chemotherapy (Chi-squre test, P = 0.024). Therefore, we did the subgroup analysis and found SMARCA2 attenuation was an unfavorable factor in patients not receiving chemotherapy rather than in those receiving chemotherapy (log rank test, P = 0.032 and 0.447, respectively). In stage II-IV cases with receiving adjuvant chemotherapy (n = 703), the multivariate Cox regression analysis using backward elimination demonstrated SMARCA2 attenuation was not an independent unfavorable parameter for overall survival (P = 0.204, HR = 1.185, 95% CI 0.912-1.539). The facts suggested patients with SMARCA2 attenuated-GC might benefit from 5-fluorouracil-based chemotherapy. However, this finding needs further validation by more well-controlled studies.

Manuscript, page 15-16, line 225-233: For adjusting the influence of chemotherapy, we further evaluated the prognostic importance of SWI/SNF status in 949 cases of stage II-IV disease (S2 Table). In 946 cases with available information, 703 cases (74%) received chemotherapy and 243 cases (26%) did not. The chemotherapeutic agents were routinely administrated postoperatively until patients declined or had contraindications. Since this cohort was retrospectively collected from earlier time, the chemotherapeutic regimens were inconsistent and, in 682 cases with available data, most patients (678, 99%) received 5-fluorouracil-based regimens in the form of single agent (415, 61%, oral or intravenous) or various combinations (263, 39%).

Manuscript, page 17-18, line 264-276: For stage II-IV cases with consideration of chemotherapy, SMARCA2 attenuation remained an unfavorable indicator for overall survival (P = 0.018, HR 1.312, 95% CI 1.048-1.643). However, the ratio of patients receiving chemotherapy were different in the SMARCA2-attenuated and SMARCA2-retained groups (85/277 = 31% versus 158/669 = 24%, P = 0.024), indicating imbalance existed in the receipt of adjuvant chemotherapy. Therefore, we did the subgroup analysis and found SMARCA2 attenuation was an unfavorable factor in patients not receiving chemotherapy rather than in those receiving chemotherapy (log rank test, P = 0.032 and 0.447, respectively). In stage II-IV cases with receiving adjuvant chemotherapy (n = 703), the multivariate Cox regression analysis using backward elimination demonstrated SMARCA2 attenuation was not an independent unfavorable parameter for overall survival (P = 0.204, HR = 1.185, 95% CI 0.912-1.539). The facts suggested patients with SMARCA2 attenuated-GC might benefit from 5-fluorouracil-based chemotherapy.

4. In Figure 3a-c figure, it would be better to display a comparison of the groups by dividing into two groups (SMARCA2-attenuated and SMARCA2-retained), rather than comparing the groups by dividing them into four groups (HET, NEG, POS, and RED).

Reply: We modified the figures by dividing GC into two groups (SMARCA2-attenuated and SMARCA2-retained). 

Manuscript, Figure 3a-c.

5. (Figure2-3) Although Figure and Legend are understandable with the careful reading of the text, it is not well presented so that the result is not easily recognized. The following is not a request but a suggestion. In figure3, it is better to use “all GC”, “non-EBV/MSI diffuse/mixed”, “all GC” and “all GC” instead of “SMARCA2”, “diffuse/mixed” “ARID1A” and “SMARCB1”. And, it is better to use “SMARCA2_HET”, “SMARCA2_NEG”, “SMARCA2_POG”, and “SMARCA2_RED” instead of “HET”, “NEG”, “POG” and “RED”. In figure3e legend, it is better to use “ARID1A-lost/heterogeneous” instead of “attenuated ARID1A expression”

Reply: We revised Figure 2-3 according to your kind suggestions. In figure 3a-c, we remove the “SMARCA2_HET”, “SMARCA2_NEG”, “SMARCA2_POG”, and “SMARCA2_RED” and use SMARCA2-attenuated and SMARCA2-intact two groups according to your comment #4. In figure 3d-e, we added ARID1A and SMARCB1 for symbol labels in the pictures and used different colors in the table of number at risk. In figure 3e legend, we modified to “ARID1A-lost/heterogeneous”.

Manuscript, page 3, line 55: ARID1A-lost/heterogeneous GCs were more aggressive in the EBV genotype (P = 0.016).

Manuscript, page 19, line 295-296: Subgroup analysis identified ARID1A-lost/heterogeneous expression was associated with unfavorable outcome in only EBV-associated gastric cancer.

Manuscript, page 23, line 367: Nevertheless, the subgroup analysis demonstrated that EBV-associated GC with ARID1A-lost/heterogeneous expression exhibited more aggressive behavior (P = 0.016)…

6. (15p) It is better to use “pT4 category” and “pN3 category” instead of “T4 stage” and “N3 stage”

Reply: We revised to “pT4 category” and “pN3 category”.

Manuscript, p17, line 253: … Lauren diffuse/mixed histotype and deeper invasion (pT4 category) …

Manuscript, p17, line 255: … lymphatic invasion (P < 0.001) with a tendency toward pN3 category (P = 0.099).

Journal Requirements:

and

Reply: We follow the PLOS ONE's style requirements.

2. In the ethics statement in the manuscript and in the online submission form, please provide additional information about the patient records/samples used in your retrospective study, including: a) whether all data were fully anonymized before you accessed them; b) the date range (month and year) during which patients' medical records/samples were accessed.

Reply: We added additional information, including all data were anonymized by symbols when we accessed them and the date range (month and year) during which patients' medical records/samples were accessed.

Manuscript, page 7, line 109-114. We enrolled 1,224 patients who received gastrectomy for GC between January 1999 and December 2007 from the archive of the Department of Anatomic Pathology at Linkou Chang Gung Memorial Hospital in Taiwan. Patient demographics, tumor characteristics and clinical outcomes were collected from the medical records and the Taiwan Cancer Registry database. Patient survival was traced through July 31, 2018. All data were anonymized by symbols when we accessed them.

3. Please provide accession numbers and/or URLs for the TCGA dataset analysed.

Reply: The URL for the TCGA dataset was download from the cBioportal website (http://download.cbioportal.org/stad_tcga_pan_can_atlas_2018.tar.gz).

Manuscript, page 10, line 170. We downloaded and analyzed clinical information, somatic variants, and CNAs for 434 STAD patients from the TCGA Pan-Cancer Atlas dataset via cBioPortal (http://download.cbioportal.org/stad_tcga_pan_can_atlas_2018.tar.gz) on June 1, 2020 [17].

4. At this time, we ask that you please provide scale bars on the microscopy images presented in Figure 1and refer to the scale bar in the corresponding Figure legend.

Reply: We added scale bars on the microscopy images presented in Figure 1 and refer to the scale bar in the corresponding Figure legend.

Manuscript, page 9, line 146: Fig 1a. Three immunohistochemical patterns of attenuated SWI/SNF subunits in gastric cancers (upper left: lost pattern, upper right: reduced pattern, lower: heterogeneous pattern; scale bar in upper left and upper right: 100 μm, scale bar in lower: 200 μm).

5. Thank you for stating in your Funding Statement:

"This work was supported by grants from the Ministry of Science and Technology (108-2320-B-182A-018, 106-2320-B-182A-011-MY3 and 105-2320-B-182A-014) and the Chang Gung Memorial Hospital (CMRPG3F2073, CMRPG5K0021, CIRPG3D0153, CMRP3C1323 and CMRPG3G0553).".

i) Please provide an amended statement that declares *all* the funding or sources of support (whether external orinternal to your organization) received during this study, as detailed online in our guide for authors at http://journals.plos.org/plosone/s/submit-now. Please also include the statement “There was no additional external funding received for this study.” in your updated Funding Statement.

ii) Please include your amended Funding Statement within your cover letter. We will change the online submission form on your behalf.

Reply: We amended the statement “all the funding or sources of support were received during this study and there was no additional external funding received for this study” and included amended Funding Statement within our cover letter. 

Manuscript, page 27, line 451-454: The funders had no role in study design, data collection and analysis, decision to publish, or preparation of the manuscript. All the funding or sources of support were received during this study and there was no additional external funding received for this study.

6. We note that you have indicated that data from this study are available upon request. PLOS only allows data to be available upon request if there are legal or ethical restrictions on sharing data publicly. For information on unacceptable data access restrictions, please see http://journals.plos.org/plosone/s/data-availability#loc-unacceptable-data-access-restrictions. In your revised cover letter, please address the following prompts:

b) If there are no restrictions, please upload the minimal anonymized data set necessary to replicate your study findings as either Supporting Information files or to a stable, public repository and provide us with the relevant URLs, DOIs, or accession numbers. Please see http://www.bmj.com/content/340/bmj.c181.long for guidelines on how to de-identify and prepare clinical data for publication. For a list of acceptable repositories, please see http://journals.plos.org/plosone/s/data-availability#loc-recommended-repositories. We will update your Data Availability statement on your behalf to reflect the information you provide.

Reply: Data cannot be shared publicly due to potentially identifying and sensitive patient information. Data are only available with approval of the Institutional Ethics Committee of the Chang Gung Memorial Hospital (irb1@cgmh.org.tw) or via the corresponding author (ctc323@cgmh.org.tw) for researchers who meet the criteria for access to confidential data.

Manuscript, page 28, line 457-461: Data cannot be shared publicly due to potentially identifying and sensitive patient information. Data are only available with approval of the Institutional Ethics Committee of the Chang Gung Memorial Hospital (irb1@cgmh.org.tw) or via the corresponding author (ctc323@cgmh.org.tw) for researchers who meet the criteria for access to confidential data.

---

## [Decision Letter · Decision Letter 1]

29 Dec 2020

The clinicopathological significance of SWI/SNF alterations in gastric cancer is associated with the molecular subtypes

PONE-D-20-31951R1

Dear Dr. Chen,

We’re pleased to inform you that your manuscript has been judged scientifically suitable for publication and will be formally accepted for publication once it meets all outstanding technical requirements.

Kind regards,

Hiromu Suzuki, M.D., Ph.D.

Academic Editor

PLOS ONE

Additional Editor Comments (optional):

Reviewers' comments:

Reviewer's Responses to Questions

**Comments to the Author**

1. If the authors have adequately addressed your comments raised in a previous round of review and you feel that this manuscript is now acceptable for publication, you may indicate that here to bypass the “Comments to the Author” section, enter your conflict of interest statement in the “Confidential to Editor” section, and submit your "Accept" recommendation.

Reviewer #1: All comments have been addressed

Reviewer #2: All comments have been addressed

2. Is the manuscript technically sound, and do the data support the conclusions?

Reviewer #1: Partly

Reviewer #2: Yes

3. Has the statistical analysis been performed appropriately and rigorously? 

Reviewer #1: Yes

Reviewer #2: Yes

4. Have the authors made all data underlying the findings in their manuscript fully available?

Reviewer #1: Yes

Reviewer #2: Yes

5. Is the manuscript presented in an intelligible fashion and written in standard English?

Reviewer #1: Yes

Reviewer #2: Yes

6. Review Comments to the Author

Reviewer #1: All comments are responded in an appropriate manner. I do not understand why EO requires >100 characters. I think it is a system error. No matter how many characters I entered it keep saying "Minimum Character Count Not Met".

Reviewer #2: The authors have adequately answered my comments, and the revised manuscript was well-written. I have no special comments.

7. PLOS authors have the option to publish the peer review history of their article (what does this mean?). If published, this will include your full peer review and any attached files.

Reviewer #1: No

Reviewer #2: No

---

## [Editor Report · Acceptance letter]

11 Jan 2021

PONE-D-20-31951R1 

The clinicopathological significance of SWI/SNF alterations in gastric cancer is associated with the molecular subtypes 

Dear Dr. Chen:

I'm pleased to inform you that your manuscript has been deemed suitable for publication in PLOS ONE. Congratulations! Your manuscript is now with our production department. 

Kind regards, 

on behalf of

Dr. Hiromu Suzuki 

Academic Editor

PLOS ONE